# Spending Thinking Time Wisely: Accelerating MCTS with Virtual Expansions

## Abstract

One of the most important AI research questions is to trade off computation versus performance, since "perfect rational" exists in theory but it is impossible to achieve in practice. Recently, Monte-Carlo tree search (MCTS) has attracted considerable attention due to the significant improvement of performance in varieties of challenging domains. However, the expensive time cost during search severely restricts its scope for applications. This paper proposes the Virtual MCTS (V-MCTS), a variant of MCTS that spends adequate amounts of time to think about different questions. Inspired by this, we propose a strategy that converges to the ground truth MCTS search results with much less computation. We give theoretical bounds of the the proposed method and evaluate the performance in $9 \times 9$ Go board games and Atari games. Experiments show that our method can achieve similar performances as the original search algorithm while requiring less than $50\%$ number of search times on average. We believe that this approach is a viable alternative for tasks with limited time and resources.

## 1 Introduction

When artificial intelligence was first studied in the 1950s, researchers seek to answer the question of what is the solution to the question if the agent were "perfect rational". The term "perfect rational" here refers to the decision made with infinite amounts of computations. However, without taking into consideration the practical computation time, one can only solve small-scale problems, since classical search algorithms usually exhibit exponential running time. Recent AI researches no longer seek to achieve "perfect rational", but instead carefully trade-off computation versus the level of rationality. People have developed computational models like "bounded optimality" to model these settings (Russell & Subramanian, 1994). The increasing level of rationality under the same computational budget has given us a lot of AI successes nowadays. Notable algorithms include the Monte-Carlo sampling algorithms, the variational inference algorithms, and using neural networks as universal function approximators (Coulom, 2006; Chaslot et al., 2008; Gelly & Silver, 2011; Silver et al., 2016; Hoffman et al., 2013).

More recently, MCTS-based RL algorithms have achieved a lot of success, mainly in board games. The most notable achievement is that AlphaGO beats Hui Fan in 2015 (Silver et al., 2016). This is the first time that a computer program beats a human professional player. After that, AlphaGo beats two top-ranking human players, Lee Sedol in 2016 and Jie Ke in 2017, the latter of which rank first worldwide at the time. Later, the MCTS-based RL algorithms are further extended to other board games, as well as the Atari video games (Schrittwieser et al., 2020). EfficientZero (Ye et al., 2021) greatly improves the sample efficiency of MCTS-based RL algorithms, shedding light on its future applications in real-world applications like robotics and self-driving.

Despite the impressive performance of MCTS-based RL algorithms, they require massive computations to train and evaluate. For example, Schrittwieser et al. (2020) used 1000 TPUs trained for 12 hours to learn the game of GO, and for a single Atari game, it needs 40 TPUs to train 12 hours. Compared to previous algorithms on the Atari games benchmark, it needs around two orders of magnitude more compute. This prohibitively large computational requirement has slowed down both the further development of MCTS-based RL algorithms, as well as its practical use.

Under the hood, MCTS-based RL algorithms are model-based methods, that imagine what the futures look like when doing different future action sequences. However, this imaging process for the

current method is not computationally efficient. For example, AlphaGo needs to look ahead 1600 game states to place a single stone. On the contrary, top human professional players can only think through around 100-200 game states per minute (Silver et al., 2016). Besides being computationally inefficient, the current MCTS algorithm deals with easy cases and hard ones with the same computational budget. On the other hand, human knows to use their time when it is most needed.

In this paper, we aim to design new algorithms that save the computational time of the MCTS-based RL methods. More specifically, we are interested in pushing the Pareto front of the rationality level - computation curve. Empirical results show that our method can achieve comparable performance while requiring less than 50% simulations to search on average.

## 2 RELATED WORK

### 2.1 MULTI-ARMED BANDIT PROBLEM

RL algorithms are always brought into the exploration and exploitation dilemma. Multi-armed bandit (MAB) problem (Berry & Fristedt, 1985; Auer et al., 2002; Lattimore & Szepesvári, 2020) is one of the most extensively studied but fundamental instances. The $K$-armed MAB problem is a sequential game with a collection of $K$ unidentified but independent reward distributions, each associated with the corresponding arms. For each round, the learner pulls an arm and receives a reward sampled from the corresponding distributions. The optimal policy of the learner for the MAB problem is to maximize the cumulative rewards obtained from the sequential decisions.

In the case where the cost of pulling arms is little, the learner is allowed to trial and error for enough times until convergence. A series of upper confidence bound (UCB) algorithms (Auer et al., 2002; Bubeck & Cesa-Bianchi, 2012) are proposed to solve the stochastic MAB problem and they have theoretical bounds. When there exist costs for each trial, pure exploration attempts to make the best use of the finite trials (Bubeck et al., 2011; Lattimore & Szepesvári, 2020). Kocsis & Szepesvári (2006) proposed UCT to adapt UCB algorithms to the tree structures, which is the basis of MCTS.

### 2.2 REINFORCEMENT LEARNING WITH MCTS

For a long time, Computer Go is regarded as a very challenging game (Bouzy & Cazenave, 2001; Cai & Wunsch, 2007). Researchers attempt to use Monte-Carlo techniques that evaluate the value of the node state through random playouts (Bouzy & Helmstetter, 2004; Gelly & Silver, 2007; 2008; Silver et al., 2016). Afterwards, UCT has generally replaced those earlier heuristic methods for Monte-Carlo tree search (MCTS). UCT algorithms (Kocsis & Szepesvári, 2006) apply UCB1 to select action at each node of the tree. Recently, MCTS-based methods (Silver et al., 2016; 2017; 2018; Schrittwieser et al., 2020) have become increasingly popular and achieved super-human performances on board games because of the strong ability to search. Modern MCTS-based RL algorithms include four stages in each search iteration, namely simulation: selection, expansion, evaluation, and backpropagation. The selection stage targets selecting a new leaf node with UCT. The expansion stage expands the selected node and updates the search tree. The evaluation stage evaluates the value of the new node. The backpropagation stage propagates the newly computed value to the nodes along the search path to obtain more accurate Q-values with Bellman backup. However, search is quite time-consuming, which prevents MCTS-based methods to be used in wider scenarios.

### 2.3 ACCELERATION OF MCTS

There are two kinds of bottlenecks in MCTS in the aspect of speed: the evaluation/selection stage of each iteration and the outer search loop. In the previous research, people attempted to evaluate the node value by random playouts to the end of the game, which makes the evaluation stage quite expensive. In addition, compared to other model-free RL methods like PPO (Schulman et al., 2017) and SAC (Haarnoja et al., 2018), MCTS-based algorithms have much larger computations due to the search loop. Therefore, a lot of works are devoted to accelerating MCTS. Some heuristic pruning methods (Gelly et al., 2006; Wang & Gelly, 2007; Sephton et al., 2014; Baier & Winands, 2014; 2018) are developed to make the selection or evaluation more effectively. Lorentz (2015) proposed early playout termination of MCTS (MCTS-EPT) to stop the random playouts early and use an evaluation function to assess win or loss, which is an improvement in the evaluation stage of normal

MCTS. And Hsueh et al. (2016) applied MCTS-EPT to the Chinese dark chess. Afterwards, MCTS-EPT similar ideas have been applied in the evaluation stage of AlphaGoZero (Silver et al., 2017) and later MCTS-based methods (Silver et al., 2018; Schrittwieser et al., 2020; Ye et al., 2021) including our baseline models. They evaluate the $Q$-values through evaluation networks instead of running playouts to the end. However, these methods focus on the specific stage of the search iteration to accelerate the MCTS. We propose Virtual MCTS, which aims to terminate the outer search iteration adaptively in MCTS under distinct circumstances without sacrificing the final policy quality.

## 3 BACKGROUND

The AlphaGo series of work (Silver et al., 2016; 2017; 2018; Schrittwieser et al., 2020) are all MCTS-based reinforcement learning algorithms. Those algorithms assume the environment transition dynamics are known or learn the environment dynamics. Based on the dynamics, they use the Monte-Carlo tree search (MCTS) as the policy improvement operator. I.e. taking in the current policy, MCTS returns a better policy with the search algorithm. The systematic search allows the MCTS-based RL algorithm to quickly improve the policy, and perform much better in the setting where a lot of reasoning is required. MCTS is the core component in the algorithms like AlphaGo.

### 3.1 MCTS

In this part, we give a brief introduction to the MCTS method implemented in reinforcement learning applications. MCTS takes in the current MDP state and runs a search algorithm guided by the current policy function. It outputs an improved policy of the current state. The improved policy is later to select an action in the environment. In the selection stage, an action will be selected by maximizing over UCB. Specifically, AlphaZero (Silver et al., 2018) and MuZero (Schrittwieser et al., 2020) are developed based on a variant of UCB, P-UCT (Rosin, 2011) and have achieved great success on board games and Atari games. The formula of P-UCT in the two methods is the Eq (1):

$$a^k = \arg\max_{a \in \mathcal{A}} Q(s,a) + P(s,a)\frac{\sqrt{\sum_b N(s,b)}}{1+N(s,a)}\left(c_1 + \log\left(\frac{\sum_b N(s,b) + c_2 + 1}{c_2}\right)\right), \quad (1)$$

where $k$ is the index of the iterative step, $\mathcal{A}$ is the action set, $Q(s,a)$ is the estimated Q-value, $P(s,a)$ is the policy prior obtained from neural networks and $N(s,a)$ is the visit counts to select the action $a$ from the state $s$. The output of MCTS is the visit count of each action of the root node. After $N$ search iterations, the final policy $\pi(s)$ is defined as the normalized root visit count distribution $\pi_N(s)$, where $\pi_k(s,a) = (N(s,a))/\sum_{b \in \mathcal{A}} N(s,b) = N(s,a)/k, a \in \mathcal{A}$. For simplification, we use $\pi_k$ in place of $\pi_k(s)$ sometimes. In our method, we propose to approximate the final policy $\pi_N(s)$ with $\hat{\pi}_k(s)$, which we name as a policy candidate, through a new expansion method and a termination rule. In this way, the number of iterations in MCTS can be reduced from $N$ to $k$.

### 3.2 COMPUTATION REQUIREMENT

Most of the computations in MCTS-based RL are in the MCTS procedure. For each action taken by MCTS, it needs $N$ times neural network evaluations, where $N$ is the number of search iterations in MCTS. Traditional RL algorithms, such as PPO (Schulman et al., 2017) or DQN (Mnih et al., 2015), only need a single neural network evaluation per action. Thus, MCTS-based RL is roughly $N$ times computationally more expansive than traditional RL algorithms.

In practice, training a single Atari game needs 12 hours of computation time on 40 TPUs (Schrittwieser et al., 2020). The computation need is roughly two orders of magnitude more than traditional RL algorithms (Schulman et al., 2017), although the final performance of MuZero is much better.

## 4 METHOD

In this paper, we propose an algorithm named Virtual MCTS to reduce the computation cost of MCTS-based RL algorithms. More concretely, we aim to push the front of the Pareto curve of the performance–computation trade-off. Intuitively, human knows when to make a quick decision and when to make a slow decision in different circumstances. It gives the ability to overcome

more difficult problems without wasting much time on easy ones. This situation-aware behavior is absent in current MCTS algorithms. We propose an MCTS algorithm variant that early terminates the search iteration in MCTS adaptively under distinct situations of states. It is consists of two components, the virtual expansion to estimate the final visit count based on the current partial tree; the termination rule that decides when to terminate based on the hardness of the current scenario.

## 4.1 TERMINATION RULE

We propose to terminate the MCTS early based on the current tree statistics. Intuitively, during the MCTS tree expansion process, if we find that recent searches do not further change the root visitation distribution, then we no longer need to search further. With this intuition in mind, we propose a simple modification to the MCTS search algorithm. Let $\pi_k(s)$ denote the root visitation distribution of the root state $s$ at MCTS expansion iteration $k$. We propose to terminate when:

$$||\pi_k(s) - \pi_{k/2}(s)||_1 < \epsilon$$

where $\epsilon$ is a tolerance hyper-parameter. Note that, in MCTS-based RL algorithms, not only the best arm matters but the other arms matter as well. It is because MCTS is used in the exploration process, and we need to make sure proper exploration happens at the non-best arms. This seems to be a heuristic rule, without any guarantees whether $\pi_k$ is close to $\pi_N$, where $N$ is the original MCTS search iteration. However, we show that under certain conditions, a bound on $||\pi_k(s) - \pi_{k/2}(s)||_1$ implies a bound on $||\pi_k(s) - \pi_N(s)||_1$.

First of all, we list some notations: $k$ is the index of the current search iteration and $N$ is the number of total search simulations, $R_t(s, a)$ is the predicted $Q$-value output at the $t$-th iteration, $N_k(s, a)$ denotes the total visit counts of the action $a$ from the state $s$ after the $k$-th iteration, the $Q$-value at $k$-th iteration is $Q_k(s, a) = \sum_{t=1}^{k} R_t(s, a)/N_k(s, a)$. The Lemma here gives a brief bound for the ranges of $Q$-values at different iterations, and the proof is attached in Appendix A.2.

**Lemma 1** $\forall a \in \mathcal{A}$, given that $R_t(s, a) \in [-1, 1]$ and the $Q_k(s, a) = \frac{\sum_{t=1}^{N_k(s,a)} R_t(s,a)}{N_k(s,a)}$, then at iteration $1 \leq k_1 < k_2 \leq N$, $Q_{k_2}(s, a) - Q_{k_1}(s, a) \leq (1 - \frac{N_{k_1}(s,a)}{N_{k_2}(s,a)})(1 - Q_{k_1}(s, a))$.

It tells that the future changes of $Q$-values with $N - k$ further searches are bounded in a small range. Through the virtual expansion that we propose in Section 4.2, we can generate a policy candidate $\hat{\pi}_k$ to approximate the final oracle policy $\pi_N$. Furthermore, Lemma 2 measures the distance between the oracle policy $\pi_N$ and our current policy candidate $\hat{\pi}_k$ and the proof is attached in Appendix A.2. If the policy candidates $\hat{\pi}_k$ and $\hat{\pi}_{k/2}$ are close enough, so are $\hat{\pi}_k$ and $\hat{\pi}_N$. In the proof, we show that $\hat{\pi}_N$ is equal to the oracle policy $\pi_N$. Therefore we conclude that the current policy candidate $\hat{\pi}_k$ is a near-oracle policy if the termination rule $||\hat{\pi}_k(s) - \hat{\pi}_{k/2}(s)||_1 < \epsilon$ is satisfied.

**Lemma 2** $\forall a \in \mathcal{A}$, given that $r \in (0, 1]$, if $\exists k \in [rN, N], ||\hat{\pi}_k(s) - \hat{\pi}_{k/2}(s)||_1 < \epsilon$, then $||\hat{\pi}_k(s) - \pi_N(s)||_1 < \epsilon + 1 - r$.

## 4.2 VIRTUAL EXPANSION IN MCTS

In the derivation above, we assume $\pi_i$ and $\pi_j$ are directly comparable. Here $\pi_i$ and $\pi_j$ denote two root node visit count distributions at iteration $i$ and $j$ respectively. However, because the tree is expanded with UCT, they are not directly comparable. UCT is an algorithm that maintains the upper bound of the node values in the search tree. As the number of visits increases, the upper bound would be tighter and the latter visits are more focused on the most promising part of the tree. Thus earlier visit count distributions (smaller iteration number) can exhibit more exploratory distribution, while latter ones (larger iteration number) are more exploitative on promising parts.

To compare $\pi_i$ and $\pi_j$ properly, we propose a method called virtual expansion in MCTS. In a nutshell, it aligns two distributions by virtual UCT expansions until the constant budget $N$. When the tree is expanded at iteration $i$, it has $N - i$ iterations to go. A normal expansion would require evaluating neural network $N - i$ times for a more accurate $Q(s, a)$ estimate for each arm at the root node. Our proposed virtual expansion still expands $N - i$ times according to UCT, but it ignores the $N - i$ neural network evaluations and simply assumes that each arm's $Q(s, a)$ does not change. We denote the virtual expanded distribution from $\pi_i$ as a policy candidate $\hat{\pi}_i$. By doing virtual expansions on both $\pi_i$ and $\pi_j$, we effectively remove the different levels of exploration/exploitation in them.

**Algorithm 1** Iteration of Search in MCTS

1: Current $k$-th iteration step:
2: Given: $\mathcal{A}, P, Q_k(s,a), N_k(s,a)$
3: $s \leftarrow s_{\text{root}}$
4: **repeat** do search
5:     $a^* \leftarrow UCT(Q, P, N)$
6:     $s \leftarrow \text{next state}(s, a^*)$
7: **until** $N_k(s, a^*) = 0$
8: Evaluate the state value $R(s,a)$ and $P(s,a)$
9: **for** $s$ along the search path **do**
10:     $Q_{k+1}(s,a) = \frac{N_k(s,a)\cdot Q_k(s,a)+R(s,a)}{N_k(s,a)+1}$
11:     $N_{k+1}(s,a) = N_k(s,a) + 1$
12: **end for**
13: **Return** $Q_{k+1}(s,a), N_{k+1}(s,a)$

**Algorithm 2** Iteration of Search in MCTS with Virtual Expansion

1: Current $k$-th iteration step:
2: Given: $\mathcal{A}, P, Q_k(s,a), N_k(s,a), \hat{N}_k(s,a)$
3:
4: **if** Not init $\hat{N}_k(s,a)$ **then**
5:     Init: $\hat{N}_k(s,a) \leftarrow N_k(s,a)$
6: **end if**
7:
8: $s \leftarrow s_{\text{root}}$
9: $a^* \leftarrow UCT(Q, P, \hat{N})$
10: $\hat{N}_k(s,a) \leftarrow \hat{N}_k(s,a) + 1$
11:
12: **Return** $\hat{N}_k(s,a)$

The comparisons between the MCTS and the one with virtual expansion are illustrated in Algorithm 1, 2. Here we display the complete one-step iteration of MCTS with or without virtual expansion. The time-consuming computations are highlighted in Algorithm 1. Line 4 to 7 in Algorithm 1 target at searching with UCT to reach an unvisited state for exploration. Then it evaluates the state and backpropagates along the search path to fit a better estimation of $Q$-values. After a total of $N$ iterations, the visit count distribution of the root node $\pi_N(s)$ is considered as the final policy $\pi(s)$. However, in the MCTS with virtual expansions, listed in Algorithm 2, it only searches one step from the root node and selects the action based on the current estimations without changing any properties of the search tree. Furthermore, the virtual visited counts $\hat{N}_k(s,a)$ are changed after virtual visits to balance the exploitation and the exploration issue. Then the policy candidate after virtual expansions becomes $\hat{\pi}_k(s,a) = \hat{N}_k(s,a)/N$ instead of $N_k(s,a)/k$. When $k = N$, further searches after the root have no effects on the final policy. So we have $\hat{\pi}_N(s,a) = \pi_N(s,a)$. In this way, the final visit count distribution obtained through the virtual expansions is similar to the oracle one.

## 4.3 V-MCTS ALGORITHM SUMMARY

The procedure of MCTS with the termination rule is listed as Algorithm 3. Compared with the original MCTS, the line 8-13 are the pseudo code on the termination rule. In each iteration, we do some calculations with little cost to judge whether the condition of termination is satisfied. If it is, then the search process is terminated and returns the current policy candidate $\hat{\pi}_k(s)$. Thus, it skips the next $N - k$ model predictions from neural networks in the evaluation stage highlighted in line 5. In this way, we can approximate the oracle distribution $\pi_N$ by $\hat{\pi}_k$ while reducing the budget of $N$ simulations to $k$. Here, $k \geq rN$ and $r$ is a hyperparameter of the minimum budget $rN$. Then we can reduce the tree size by $1/r$ times at most. However, $r$ cannot be a tiny

**Algorithm 3** Virtual MCTS

1: Given budget $N$, state $s$, conservativeness $r$, error $\epsilon$
2: **for** $k \in N$ **do**
3:     Selection with UCT
4:     Expansion for the new node
5:     Evaluation with Neural Networks
6:     Backpropagation for updating Q and visit counts
7:     $\pi_k(s,a) \leftarrow N_k(s,a)/n$
8:     Virtual expand $N - k$ nodes and update $\hat{N}(s,a)$
9:     $\hat{\pi}_k(s) \leftarrow \hat{N}(s,a)/N$
10:     **if** $k \geq rN \wedge \left\|\hat{\pi}_k(s) - \hat{\pi}_{k/2}(s)\right\|_1 < \epsilon$ **then**
11:        $\pi(s) \leftarrow \hat{\pi}_k(s)$
12:        **Break**
13:     **end if**
14:     $\pi(s) \leftarrow \pi_k(s)$
15: **end for**
16: **Return** $\pi(s)$

value as the minimum distance of distributions is bounded by $r$. Otherwise, the termination rule has little effect. In conclusion, this rule tells the MCTS to terminate if the policy candidates have converged. On such occasions, the virtual expansion method has similar effects as the real expansion of MCTS to generate a near-oracle policy, which can save the time cost of the left $N - k$ simulations.

Furthermore, in the next section, we do some ablations to further investigate the effects of $r, \epsilon$ and visualizations to understand why the termination rule works. We name our method Virtual MCTS (V-MCTS), a variant of MCTS with a termination rule based on the virtual expansion.

## 5 EXPERIMENTS

In this section, the goal of the experiments is to prove the effectiveness and efficiency of our proposed algorithm. We compare the performance as well as the cost of the budget of the MCTS-based methods with or without the termination rule. Specifically, we evaluate the board game Go $9 \times 9$, and a few Atari games. In addition, we do some ablations to further examine the effectiveness of the virtual expansion and evaluate how sensitive our method is to the hyper-parameters. Finally, we try to understand the adaptive mechanism with visualizations and performance analysis.

### 5.1 SETUP

Recently, Ye et al. (2021) proposed EfficientZero, a variant of MuZero (Schrittwieser et al., 2020) with three extra components to improve the sample efficiency, which only requires 8 GPUs in training, and thus it is more affordable. Here we choose the board game Go $9 \times 9$ as our benchmark environment. The game of Go tests how the algorithm performs in a challenging planning problem. We also benchmark on a few Atari games, which feature complexity on the visual side.

As for the Go $9 \times 9$, we choose Tromp-Taylor rules during training and evaluation. The environment of Go is built based on an open-source codebase, GymGo (Huang, 2021). We evaluate the performance of the agent against GNU Go v3.8 at level 10 (Bump et al., 2005) for 200 games. We include 100 games as the black player and 100 games as the white one with different seeds. We set the komi to 6.5 as most papers do. As for the Atari games, we choose 5 games with 100k environment steps, which follows the setting of EfficientZero. We evaluate all these games for 32 distinct seeds.

### 5.2 COMPARATIVE EVALUATION

We compare our method to EfficientZero with original MCTS, on Go $9 \times 9$ and some Atari games. Figure 1a illustrates the comparisons on Go among different algorithms or models against the GnuGo (level 10) agent. The x-axis is the training speed and the y-axis is the winning rate. Here, we train the baseline method with different constant budgets $N$, noted as the blue points. Besides, we also train the V-MCTS with hyperparameters $r = 0.2, \epsilon = 0.1$. We evaluate the trained model with different $\epsilon$ to display the tradeoff between performance and the time cost, noted as the red points. The pink points are the GnuGo with different levels.

The result shows that the performance of V-MCTS is comparable to the MCTS of maximum budget ($N = 150$) while it requires less time to search. Notably, V-MCTS achieves a 72% winning rate against the GnuGo level 10. Meanwhile, the time cost of our method for a one-step move is 0.12s while the GnuGo engine is 0.18s. For the data points with the winning rate higher than 50%, the V-MCTS is significantly better than the original MCTS considering both the winning rate and the time cost. Consequently, we believe that such termination rule of MCTS can keep the strong performance while saving lots of budgets, which means our method is effective enough and more time-efficient.

The training curve presented in Figure 1b illustrates that the budget of search times is quite significant to the MCTS method. However, the performance of V-MCTS is better than that of MCTS with $N = 120$ while maintaining an average tree size of less than 80. Furthermore, it is interesting that the tree size varies over training procedures. In the beginning, the outputs of the value network are close to zero because the agent usually draws in self-play and receives the zero reward signal, which means the little changes of $Q_k(s, a)$ during the search. The visit count distribution of the root after virtual expansions is similar to that after real expansions. Therefore, in this stage, the termination condition is easy to meet. Afterward, as the model is trained better and receives more diverse reward signals during self-play, the probability of searching some valuable states becomes much larger. The value $Q_k(s, a)$ varies considerably in each iteration of the search, which results in the changes of distributions of policy candidates. Thus, it is much more difficult for the search process to terminate and leads to a larger number of tree sizes. Finally, with more training steps, the prediction of the policy network is more accurate and then gives a stronger prior heuristic knowledge $P(s, a)$ before

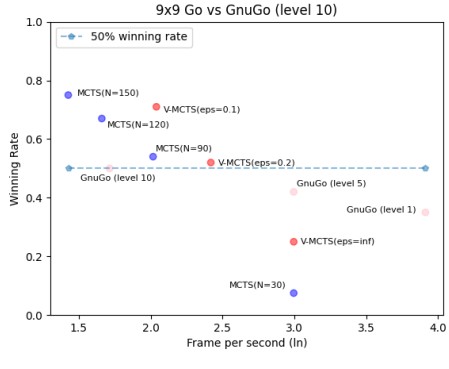

(a) Evaluations of Performance

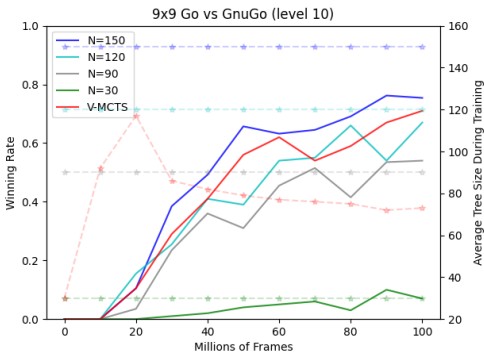

(b) Wining Rates and Tree Size on Training Stage

Figure 1: Performance of Virtual MCTS on Go $9 \times 9$ against GnuGo (level 10). (a) Evaluating the speed of search and winning rate of MCTS, V-MCTS as well as the GnuGo at different levels. The termination rule is able to reduce the search cost while keeping comparable performance and it outperforms the GnuGo at level 10 in the aspect of speed and winning rate. (b) Evaluating the winning rate as well as the average tree size in different training phases. The solid lines and dashed lines display the winning probability and the tree size respectively. The dark blue curve is the oracle version of MCTS ($N = 150$) and the red one is the version of V-MCTS. The termination rule can make the tree size adaptive in training with a little loss of performance. Directly reducing $N$ in MCTS (the grey curve with $N = 90$) results in a larger performance drop.

Table 1: Results from Atari games: scores over total 32 seeds on 5 environments. Here MCTS ($N = 50$) is the oracle version and the best results among those versions except the oracle are marked in bold. V-MCTS outperforms MCTS ($N = 30$) while requiring fewer search times.

|  | **MCTS** ($N = 50$) | **MCTS** ($N = 30$) | **MCTS** ($N = 10$) | **Ours** | **Size Avg.** |
|---|---|---|---|---|---|
| Pong | 20.8 | 17.2 | 2.1 | **19.9** | 13 |
| Breakout | 411.1 | 347.4 | 309.0 | **389.2** | 16 |
| Seaquest | 1737.5 | 930.6 | 625.6 | **1340.1** | 15 |
| Hero | 9715.0 | **7499.1** | 7310.0 | 7465.0 | 15 |
| Qbert | 15465.6 | 7792.9 | 6035.2 | **10880.5** | 17 |

search. For those actions with higher prior knowledge, the changes of $Q_k(s, a)$ have less impact on the UCT scores. The procedure of the virtual expansion is similar to the real expansion. However, the changes of $Q_k(s, a)$ are still possible during the search process. Consequently, the average number of tree size keeps in a reasonable range, which is larger than that in the innocent beginning stage and the minimum budget $rN$. Our method can determine whether to continue searching or not.

Apart from the results of Go, we also evaluate our method on some visually complex games. Since the search space of Atari games is much smaller than that of Go and the Atari games are easier, here we choose a few Atari games to study how the proposed method impacts the performance under less necessity of search. We follow the setting of EffcientZero, 100k Atari benchmark, which contains only 400k frames data. The results are shown in Table 1. Generally, we find that our method works in Atari games. The tree size is adaptive and the performance of V-MCTS is still comparable to the MCTS with full search trails. It has better performance than the MCTS($N = 30$) while requiring much fewer searches, which proves the effectiveness and the efficiency of the termination rule in the tree search method. The Hero game is not an outlier considering the similar performance between $N = 50$ and $N = 30$. Besides, the number of search times decreases more than that on Go.

To sum up, Virtual MCTS shapes better policy candidates close to the oracles through the virtual expansion with less cost of the budget. The performance of V-MCTS can keep sound with fewer search iterations while simply reducing the total budget of MCTS will encounter a larger performance drop. In addition, the savings of search cost is more substantial in easier environments.

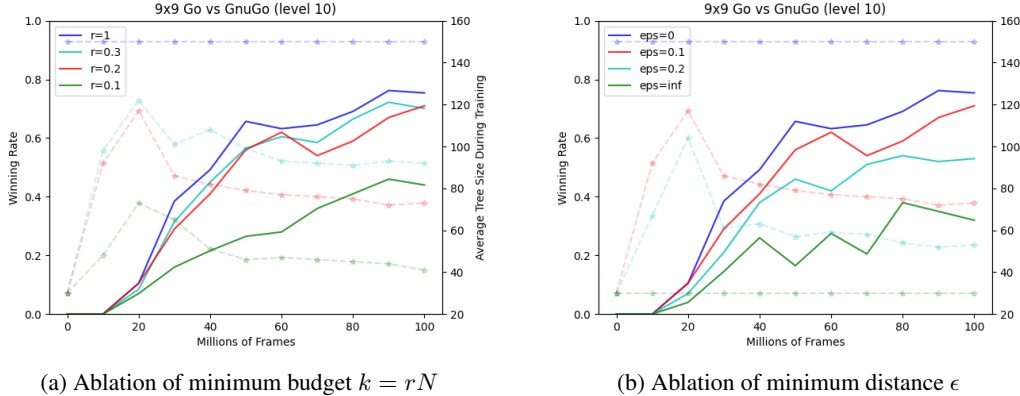

(a) Ablation of minimum budget $k = rN$   (b) Ablation of minimum distance $\epsilon$

Figure 2: Sensitivity of the termination rules to the hyperparameter $r, \epsilon$ on Go $9 \times 9$. The solid lines and dashed lines display the winning probability and the average tree size respectively.

Table 2: Ablation results of different expansion methods on Go $9 \times 9$ against GnuGo (level 10).

| Algorithm | Size Avg. | Winning Rate |
|---|---|---|
| Original expansion | 30 | 16% |
| Greedy expansion | 30 | 5% |
| Virtual expansion | 30 | 32% |

## 5.3 ABLATION STUDY

The results in the previous section suggest that our method reduces the response time of MCTS while keeping considerable performance on challenging tasks. In this section, we try to figure out which component contributes to the performance and how the hyperparameters affect it.

**Virtual Expansion** In Section 4.2, we introduce the virtual expansion and discuss the difference between the MCTS with and without virtual expansion. We compare the MCTS with virtual expansion and another two expansion methods. Here we introduce the two methods briefly. One is the original expansion, which does nothing once termination. It samples an action directly from $\pi_k(s, a) = N_k(s, a)/k$. Another is the greedy expansion, which spends the left $N - k$ simulations in searching the current best action greedily, indicating that $\hat{\pi}_k(s, a) = (N_k(s, a) + (N - k)\mathbb{I}_{a = \arg\max N_k(s, a)})/N$. Besides, we stop the search process after $k$ iterations regardless of the termination condition, where $k = rN$ and $r = 0.2, N = 150$.

We compare the winning rate against the GnuGo engine and the results are listed as Table 2 shows. All the versions here only search for the given minimum tree size, and the virtual expansion method can still achieve a 32% winning rate, which is much better than the others. Notably, MCTS with greedy expansion does not work. It is over-exploitation and results in severe exploration issues. Consequently, the virtual expansion method can generate a better policy distribution because it can balance the exploration and exploitation problem through UCT with further virtual simulations.

**Termination Rule** Since the virtual expansion provides a better choice of policy distributions, it is significant to explore a better termination rule to keep the sound performance while decreasing the tree size as much as possible. As mentioned in Section 4.1, the termination rule sets two hyperparameters $r, \epsilon$ to determine the termination rule. Then we do ablations for the different values of $r$ and $\epsilon$ respectively. The default values of $r, \epsilon$ are set to $0.2, 0.1$ in all experiments here.

Figure 2 compares the winning rate as well as the average tree size across the training stage. Firstly, Figure 2a gives the results of different minimum search times $r$. The winning probability is not sensitive to $r$ when $r \geq 0.2$. But the average tree size is sensitive to $r$ because V-MCTS is supposed to search for at least $rN$ times. In addition, there is a drop between the performance between $r = 0.1$ and $r = 0.2$. Therefore, it is reasonable to choose $r = 0.2$ to balance the speed and the performance.

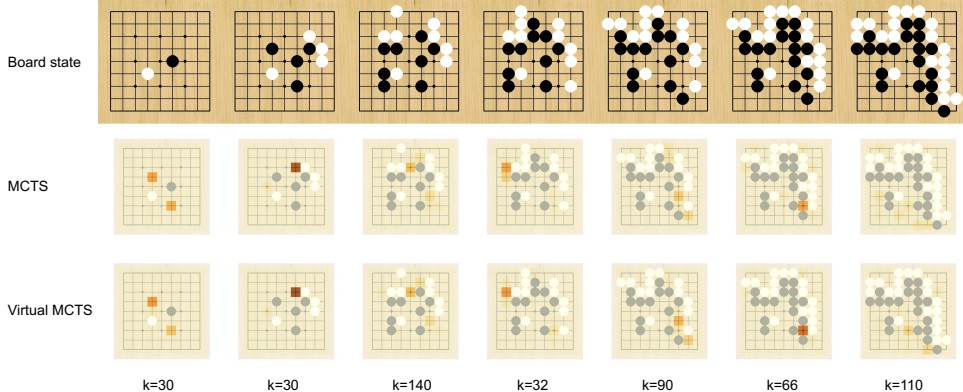

Figure 3: Heatmap of policy distributions from the MCTS ($N = 150$) and the V-MCTS. A darker red color represents larger visit counts of the corresponding action. The V-MCTS will terminate with different search times $k$ according to the situations and generate a near-oracle policy distribution.

Besides, the comparisons of different minimum distance $\epsilon$ are shown in Figure 2b. A larger $\epsilon$ makes the tree size smaller because $\left\|\hat{\pi}_k(s) - \hat{\pi}_{k/2}(s)\right\|_1 < \epsilon$ is easier to reach. In practical, we find that the performance is highly correlated with $\epsilon$. In terms of the winning probability, a smaller $\epsilon$ outperforms a larger one. However, the better performance is at cost of a larger response time. Therefore, it is a good choice to set $r = 0.2, \epsilon = 0.1$. We suggest selecting an appropriate minimum distance to balance the performance and the response time.

## 5.4 VISUALIZATION

In this section, we do some visualizations to better understand the behavior of Virtual MCTS. Specifically, we choose some states at different time steps on one game of Go against the GnuGo with a trained model and visualize the heatmap of policy distribution, as Figure 3 shows. In this figure, our player is the black one. The board states are shown in the first row, and the next two rows are the heatmap visualization for oracle MCTS and V-MCTS. The darker the color is on the grid, the more the corresponding action is visited during the search. In general, $\hat{\pi}_k$ is close to the $\pi_N$ at distinct states, indicating that the termination rule is reasonable and effective. The less valuable actions there are, the sooner the V-MCTS will terminate. For example, on Go games, the start states are usually not complex because there are only a few stones on the board but the situations are much more complicated in shuban, the closing stage of the game. Notably, the termination occurs earlier in the start states (columns 1, 2) but it is the opposite when the situation is more complicated. More importantly, the termination step $k$ is not related to the number of Go pieces. And the policy candidate obtained after virtual expansion can be close to the oracle one at distinct states of a game. Therefore, we can conclude that V-MCTS makes adaptive terminations according to the situations of the current state, which is the key to maintaining comparable performances.

## 6 DISCUSSION

In this paper, we propose a novel method named V-MCTS to accelerate the MCTS to determine the termination of search iterations. It can keep comparable performances while reducing half of the time to search adaptively. We are in the belief that this work can be one step toward applying the MCTS-based methods to some real-time domains.

## 7   REPRODUCIBILITY STATEMENT

The main implementations of our proposed method are in Algorithm 2 and 3. In addition, the settings of the experiments and hyper-parameters we choose are in Appendix A.1. The proof of the lemma is in Appendix A.2. More significantly, the details of the design of training procedures for Go are around Appendix A.1.2. Besides, we will release the codebase if this paper is accepted.

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

## A    APPENDIX

### A.1    EXPERIMENTAL SETUP

#### A.1.1    MODELS AND HYPER-PARAMETERS

**Model Design**    As for the architecture of the networks, we follow the implementation of EfficientZero (Ye et al., 2021) in Atari games, which proposes three components based on MuZero:

self-supervised consistency, value prefix, and off-policy correction. In the implementation of EfficientZero, there are a representation network, a dynamics network and a reward/value/policy prediction network. The representation network is to encode observations to hidden states. The dynamics network is to predict the next hidden state given the current hidden state and an action. The reward/value/policy prediction network is to predict the reward/value/policy. Notably, they propose to keep temporal consistency between $s_{t+1}$ and the predicted state $\hat{s}_{t+1}$. The training objective is:

$$\mathcal{L}_t(\theta) = \lambda_1 \mathcal{L}(u_t, r_t) + \lambda_2 \mathcal{L}(\pi_t, p_t) + \lambda_3 \mathcal{L}(z_t, v_t) + \lambda_4 \mathcal{L}_{\text{similarity}}(s_{t+1}, \hat{s}_{t+1}) + c||\theta||^2$$

$$\mathcal{L}(\theta) = \frac{1}{l_{\text{unroll}}} \sum_{i=0}^{l_{\text{unroll}}-1} \mathcal{L}_{t+i}(\theta), \tag{2}$$

where $u_t, \pi_t, z_t$ are the target reward/policy/value of the state $s_t$ and $r_t, p_t, v_t$ are the predicted reward/policy/value of the state $s_t$ respectively. The prediction will do $l_{\text{unroll}} = 5$ times iteratively for the state $s$ on both Go and Atari games. We do some changes when dealing with board games. For example, we remove the reward prediction network because the agent will receive a reward only at the end of the games. The other major changes for board games are listed as follows.

Since the board game Go is harder than the Atari games, we add more residual blocks (two times blocks). Specifically, we use 2 residual blocks in the representation network, the dynamics network as well as the value/policy prediction network on Go $9 \times 9$ while EfficientZero uses only 1 residual block in those networks on Atari games. As for the representation network, we remove the downsampling part here because there is no need to do downsampling for Go states. In the value/policy prediction networks, we enlarge the dimension of the hidden layer from 32 to 128. Besides, considering that the reward is sparse on Go (only the final value) and the collected data are sufficient, we only take the self-supervised consistency component in EfficientZero to give more temporal supervision during training.

**Hyper-parameters** In each case, we train EfficientZero for unrolled 5 steps and mini-batches of size 256. Besides, the model is trained for 100k batches with 100M frames of data in board games while 100k batches with 400k frames in Atari games. We stack 8 frames in board games without frameskip while stacking 4 frames in Atari games with a frameskip of 4. During both training and evaluation, EfficientZero chooses 150 simulations of budget for each search in board games while 50 simulations of budget in Atari games. Other hyper-parameters are listed in Table 3.

| | **Go** $9 \times 9$ | **Atari** |
|---|---|---|
| Maximum number of tree size | 150 | 50 |
| Observation down-sampling | No | $96 \times 96$ |
| Total frames | 100M | 400k |
| Replay buffer size | 2M | 100k |
| Max frames per episode | 163 | 108k |
| Cost of training time | 24h | 8h |
| Komi of Go | 6.5 | - |
| Frame stack | 8 | 4 |
| Frame skip | 1 | 4 |
| Training steps | 100k | 100k |
| Mini batch | 256 | 256 |
| Learning rate | 0.05 | 0.2 |
| Weight decay ($c$) | 0.0001 | 0.0001 |
| Reward loss coefficient ($\lambda_1$) | 0 | 1 |
| Policy loss coefficient ($\lambda_2$) | 1 | 1 |
| Value loss coefficient ($\lambda_3$) | 1 | 0.25 |
| Consistency loss coefficient ($\lambda_4$) | 2 | 0.5 |
| Dirichlet $\alpha$ | 0.03 | 0.3 |
| $c_1$ in P-UCT | 1.25 | 1.25 |
| $c_2$ in P-UCT | 19652 | 19652 |
| $\epsilon$ | 0.1 | 0.1 |
| $r$ | 0.2 | 0.2 |

Table 3: Hyper-parameters of V-MCTS on Go $9 \times 9$ and Atari games

### A.1.2 TRAINING DETAILS OF GO

The detailed implementations of Atari games are discussed in EfficientZero (Ye et al., 2021). However, it is nontrivial to adapt to board games. Here we give detailed instructions for training board games Go $9 \times 9$ in our implementations.

**Inputs** We follow the designs of AlphaZero and we use the Tromp-Taylor rules, which is similar to previous work (Silver et al., 2018; Schrittwieser et al., 2020). The input states of the Go board are encoded into a $17 \times 9 \times 9$ array which stacks the historical 8 frames and uses the last channels $C$ to identify the current player, 0 for black and 1 for white. Notably, the one historical frame consists of two planes $[X, Y]$, where the first plane $X$ represents the stones of the current player and the second one $[Y]$ represents the stones of the opponent. Besides, if there is a stone on board, then the state of the corresponding position in the frame will be set to 1, otherwise to 0. For example, if the current player is black and suppose $b[i, j]$ is the current board state, $X[i, j] = \mathbf{1}_{b[i,j]=\text{black stone}}, Y[i, j] = \mathbf{1}_{b[i,j]=\text{white stone}}$. In summary, we concatenate together the historical planes to generate the input state $s = [X_{t-7}, Y_{t-7}, X_{t-6}, X_{t-6}, ..., X_t, Y_t, C]$, where $X_t, Y_t$ are the feature planes at time step $t$ and $C$ gives information of the current player.

**Training** As for the training phase, we train the model from scratch without any human expert data, which is the same as the setting of Atari games. Besides, limited to the GPU resources, we do not use the reanalyzing mechanism of MuZero (Schrittwieser et al., 2020) and EfficientZero (Ye et al., 2021), which targets at recalculation of the target values and policies from trajectories in the replay buffer with the current fresher model. Specifically, we use 6 GPUs for doing self-play to collect data, 1 GPU for training, and 1 GPU for evaluation.

**Exploration** To make a better exploration on Go, we reduce the $\alpha$ in the Dirichlet noise $\text{Dir}(\alpha)$ from 0.3 to 0.03 and we scale the exploration noise through the typical number of legal actions, which follows these works (Silver et al., 2018; Schrittwieser et al., 2020). In terms of sampling actions from MCTS visit distributions, we will mask the MCTS visit distributions with the legal actions and sample an action $a_t$, where

$$a_t := \begin{cases} a_t \sim \pi_t, & t < T \\ a_t = \arg\max \pi_t, & t \geq T \end{cases} \tag{3}$$

$T$ is set to 16 in self-play and is set to 0 in evaluation, which is similar to these works (Silver et al., 2018; Schrittwieser et al., 2020). In this way, the agent does more explorations for the previous $T$ steps while taking the best action afterwards. But for Atari games, $\forall t$, we choose $a_t \sim \pi_t$ in self-play and $a_t = \arg\max \pi_t$ in evaluation, which is the same as these works (Schrittwieser et al., 2020; Ye et al., 2021).

**Two-player MCTS** On board games, there are two players against each other, which is different from that of one-player games. Therefore, we should do some changes to the MCTS with the two-player game. For one thing, the value network always predicts the Q-value of the black player instead of the current player, which provides a more stable prediction. Furthermore, A significant change is that during backpropagation of MCTS, the value should be updated with the negative value from the child node. Because the child node is the opponent, the higher value of the opponent indicates a worse value of the current player. Besides, as for $Q$-values of the unvisited children on Go and Atari games, we follow the implementation of EfficientZero (Ye et al., 2021) as follows:

$$\hat{Q}(s^{\text{root}}) = 0$$
$$\hat{Q}(s) = \frac{\hat{Q}(s^{\text{parent}}) + \sum_b \mathbf{1}_{N(s,b)>0} Q(s, b)}{1 + \sum_b \mathbf{1}_{N(s,b)>0}}$$
$$Q(s, a) := \begin{cases} Q(s, a) & N(s, a) > 0 \\ \hat{Q}(s) & N(s, a) = 0 \end{cases} \tag{4}$$

Notably, we allow the resignation for players when $\max_{a \in \mathcal{A}} Q(s^{root}, a) < -0.9$ during self-play and evaluation, which means that the predicted winning probability is less than 5%. As for other hyperparameters on both Go and Atari games, we note that we choose the same values as those in EffcientZero. Specifically, the $c_1, c_2$ in our mentioned P-UCT formula (Eq. 1) are set to 1.25 and 19652, following these works (Schrittwieser et al., 2020; Ye et al., 2021).

## A.2 PROOF

**Lemma 1**. $\forall a \in \mathcal{A}$, given that $R_t(s,a) \in [-1,1]$ and the $Q_k(s,a) = \frac{\sum_{t=1}^{N_k(s,a)} R_t(s,a)}{N_k(s,a)}$, then at iteration $1 \le k_1 < k_2 \le N$, $Q_{k_2}(s,a) - Q_{k_1}(s,a) \le (1 - \frac{N_{k_1}(s,a)}{N_{k_2}(s,a)})(1 - Q_{k_1}(s,a))$.

*proof.* Since $Q_k(s,a) = \frac{\sum_{t=1}^{k} R_t(s,a)}{N_k(s,a)}$, we have

$$Q_{k_2}(s,a) = \frac{N_{k_1}(s,a)Q_{k_1}(s,a) + \sum_{t=N_{k_1}(s,a)}^{N_{k_2}(s,a)} R_t(s,a)}{N_{k_2}(s,a)} \tag{5}$$

Because $R_t(s,a) \in [-1,1]$, then

$$Q_{k_2}(s,a) \le \frac{N_{k_1}(s,a)}{N_{k_2}(s,a)}Q_{k_1}(s,a) + \frac{N_{k_2}(s,a) - N_{k_1}(s,a)}{N_{k_2}(s,a)} \tag{6}$$

Therefore

$$Q_{k_2}(s,a) - Q_{k_1}(s,a) \le (1 - \frac{N_{k_1}(s,a)}{N_{k_2}(s,a)})(1 - Q_{k_1}(s,a)) \tag{7}$$

**Lemma 2**. Given that $r \in (0,1]$, if $\exists k \in [rN, N], \left|\left|\hat{\pi}_k(s) - \hat{\pi}_{k/2}(s)\right|\right|_1 < \epsilon$, then $\left|\left|\hat{\pi}_k(s) - \pi_N(s)\right|\right|_1 < \epsilon + 1 - r$

*proof.* Since the $\hat{\pi}_k(s,a)$ is a distribution that equals to $\frac{\hat{N}_k(s,a)}{N}$. Therefore,

$$\left|\left|\hat{\pi}_N(s) - \hat{\pi}_k(s)\right|\right|_1 \le \left|\left|\hat{\pi}_N(s) - \hat{\pi}_{N/2}(s)\right|\right|_1 + \left|\left|\hat{\pi}_{N/2}(s) - \hat{\pi}_k(s)\right|\right|_1 \tag{8}$$

Then

$$\left|\left|\hat{\pi}_N(s) - \hat{\pi}_k(s)\right|\right|_1 \le \sum_{t=0}^{M} \left|\left|\hat{\pi}_{\frac{N}{2^t}}(s) - \hat{\pi}_{\frac{N}{2^{t+1}}}(s)\right|\right|_1 + \left|\left|\hat{\pi}_k(s) - \hat{\pi}_{\frac{N}{2^{M+1}}}(s)\right|\right|_1 \tag{9}$$

, where $\frac{N}{2^{M+2}} \le k \le \frac{N}{2^{M+1}}$. In addition, $\left|\left|\hat{\pi}_k(s) - \hat{\pi}_{\frac{N}{2^{M+1}}}(s)\right|\right|_1 \le \left|\left|\hat{\pi}_k(s) - \hat{\pi}_{k/2}(s)\right|\right|_1 < \epsilon$. Besides, $k \ge rN$, then we have

$$
\begin{aligned}
\left|\left|\hat{\pi}_N(s) - \hat{\pi}_k(s)\right|\right|_1 &\le \sum_{t=0}^{M} \left|\left|\hat{\pi}_{\frac{N}{2^t}}(s) - \hat{\pi}_{\frac{N}{2^{t+1}}}(s)\right|\right|_1 + \left|\left|\hat{\pi}_k(s) - \hat{\pi}_{\frac{N}{2^{M+1}}}(s)\right|\right|_1 \\
&< \sum_{t=0}^{M} \left|\left|\hat{\pi}_{\frac{N}{2^t}}(s) - \hat{\pi}_{\frac{N}{2^{t+1}}}(s)\right|\right|_1 + \epsilon \\
&\le 1 + \epsilon - \frac{1}{2^M} \\
&\le \epsilon + 1 - r
\end{aligned}
\tag{10}
$$

For the $N$-th iteration of the search process, the final visit count distributions keep the same between the original expansion (Algorithm 1) and the virtual expansion (Algorithm 2). This is because at the last iteration, searching the nodes after the root has no effects on the final distribution. Therefore, $\hat{\pi}_N(s) = \pi_N(s)$. So we have

$$\left|\left|\hat{\pi}_k(s) - \pi_N(s)\right|\right|_1 < \epsilon + 1 - r \tag{11}$$

