# OpenReview forum: "Spending Thinking Time Wisely: Accelerating MCTS with Virtual Expansions"
_ICLR.cc/2022/Conference — ICLR 2022 Submitted_

### Official Review · Reviewer_d2cw · 2021-10-29

**Correctness:** 3
**Technical Novelty And Significance:** 3
**Empirical Novelty And Significance:** 3
**Recommendation:** 8
**Confidence:** 4

**Main Review:**

**Primary strengths**:
1) Solid and extensive empirical evaluation with ablation studies.
2) The main ideas of the paper are clearly described.
3) The proposed approach looks interesting and useful. I especially like the idea of virtual expansions, even if (or maybe especially because) it is very simple but effective. It would be even more elegant if the changes in distributions from steps $k$ to $N$ could be computed at once, rather than in $N - k$ little steps... but I suppose that might be impossible. In practice it probably doesn't matter much, it should be really fast anyway without actually rolling out any new game states or performing any neural network evaluations.

**Primary weaknesses**:
1) There are a few technical errors (relatively easily fixable ones, but also important ones).

    Firstly, section 3.1 describes Eq. (1) as "selection rule named UCT". This is wrong in several ways. UCT is actually a variant of the entire MCTS algorithm (not a selection rule), which uses *UCB1* as its selection rule. The selection rule described by Eq. (1) is a different one; it's the PUCT variant used by AlphaGo/AlphaGo Zero/AlphaZero/etc. That particular slowly-decreasing exploration hyperparameter was only seen (if I recall correctly) in AlphaZero, before that it was just a constant.

    Secondly, there seems to be a mistake in Lemma 1: if I'm not mistaken, the $(Q_{k_1}(s, a) + 1)$ term at the end should actually be $(1 - Q_{k_1}(s, a))$. In the end this may not end up mattering too much; the extreme values that the bound can take end up being the same, but flipped around for the two different extreme values that $Q_{k_1}(s, a)$ can take, because the $Q$-values are in $[-1, 1]$... but it still looks like a mistake.

2) More small mistakes (like sloppy notation a few times, see detailed comments below) and minor grammatical/spelling issues. Overall the latter don't impact comprehensibility too much though in my opinion.

---

**Detailed comments**:
- shielding light --> shedding light
- "We propose an MCTS algorithm variant that can behave like a human." --> this kind of claims about "human-like" behaviour is always very dangerous in my opinion.
- I found the notation where $\hat{Q}$-values are individual outcomes from individual iterations, and $Q$-values are averages of multiple such $\hat{Q}$-values, confusing. They're too similar, and often in other work the hat actually signifies an "approximation" whereas the version without a hat signifies a "theoretical" or "true" value. I would recommend changing the $\hat{Q}$-values to $R$-values for example (since they're like rewards in reinforcement learning).
- The text in the main paper around Lemmas 1 and 2 should at least mention that their proofs can be found in supplementary material.
- Section ?? at end of page 4.
- Start of section 4.3 points to "line 5-10" in Algorithm 3, but I think it should actually be lines 8-13?
- Second paragraph of 5.1 mentions an agent named "GNU", but it's actually named "GNU Go"
- What do you mean by "100 pieces as the black player and 100 pieces as the white one"? Should this be "games" instead of "pieces"?
- The colours used in the plots are probably not colour-blind friendly (consider that the most common form of colour-blindness is red-green colour-blindness).
- I don't understand why some results in Table 1 are printed in boldface and others not. Every single row actually has the best result not in boldface, but.... the second best?
- (Coulom, 2006) --> (Coulom, 2007)
- Capitalise "Nature" in (Silver et al., 2016)
- Should cite the publication in Science instead of the arXiv one for AlphaZero
- Descriptions of implementation details of MCTS / experiments in supplementary material are not sufficient. "All the other implementations follows the Alpha-series paper (Silver et al., 2016; 2017; Schrittwieser, 2020)" does not tell us a lot. There are many differences between those three cited papers (AlphaGo used human expert data, playouts, REINFORCE, etc.; AlphaGo Zero used no human expert data, no playouts, MCTS visit distributions as targets for policy, etc.), and many details on MCTS-related hyperparameters are also not sufficiently described in those papers; $C_{puct}$ value, $Q$-value assigned to unvisited children, etc.
- In proof for Lemma 1, denominator in Eq. (3) should be $N_{k_2}(s, a)$ instead of just $N_{k_2}$.
- In proof for Lemma 2, Eq. (5) has a $\tilde{\pi}_N(N/2)$ term, but $N/2$ is not a state.

---

**After response from authors:** I am satisfied with how the authors have addressed the issues and updated my score accordingly.

**Summary Of The Paper:**

This paper proposes an approach for a significant speedup of Monte-Carlo Tree Search (MCTS) at a relatively small cost in playing strength. The basic idea is that, when the change in the distribution of visit counts between two different time points $\frac{k}{2}$ and $k$ is less than some constant $\epsilon$, it can also be shown that the remaining change in distributions between $k$ and $N$ (where $N$ is the maximum visit count allowed by some budget) will be bounded below some value, and if we consider such a maximum possible error to be sufficiently small we can just terminate the search early at time $k$. The paper also proposes a very simple but important idea called Virtual Expansions, which basically consists of running $N - k$ additional iterations of the bandit algorithm used by MCTS solely in the root node, providing the current estimated average values as rewards for every pull (i.e., leaving average reward estimates unchanged), in order to transform the distribution of visits at iteration $k$ into a prediction of the distribution we would end up with after $N$ iterations.

Several empirical evaluations in Go ($9\times9$ board) and a few Atari games, including ablation studies, demonstrate that the approach can substantially reduce the search times and self-play training times, while only decreasing playing strength by a small amount.

**Summary Of The Review:**

While the paper has some interesting and well-evaluated ideas, and definitely should be publishable at some point, in its current form I feel that it is not yet ready; there are too many little (but sometimes important) errors.

---

**After response from authors:** I am satisfied with how the authors have addressed the issues and updated my score accordingly.

---

> ### Author Response · Authors · 2021-11-16
> **Response to Reviewer d2cw**
>
> Thank you for your comments and advice! We hope the following address your concerns:
>
> As for the first weakness, we acknowledge that you point out the technical errors we made. We admit that we should refer to P-UCT here and add more descriptions in the related works. Thank you for your correction.
>
> As for the mistake in Lemma 1, we agree that it should be $(1 - Q_{k_1}(s, a))$ and we have corrected Lemma 1 in the revised version.
>
> As for the other detailed comments you mentioned, we have corrected them in the revised version.
>
> Here are the answers to your questions and suggestions in the "detailed comments" part and we have added detailed descriptions and corrections in the revised version concerning these questions.
>
> > Suggestion: "I would recommend changing the $\hat{Q}$-values to $R$-values for example".
>
> * We agree to use $R$-values here to give a clearer notation and we have committed the changes in the revised version.
>
> > Question: "Start of section 4.3 points to 'line 5-10' in Algorithm 3, but I think it should actually be lines 8-13?"
>
> * The answer is yes, we make a mistake here and it should be lines 8-13 and we have committed the changes in the revised version.
>
> > Question: "What do you mean by '100 pieces as the black player and 100 pieces as the white one'? Should this be 'games' instead of 'pieces'?"
>
> * It is a typo and it should be "games" here and we have committed the changes in the revised version.
>
> > Suggestion: "The colours used in the plots are probably not colour-blind friendly (consider that the most common form of colour-blindness is red-green colour-blindness)."
>
> * We will choose a more friendly colour in the final version.
>
> >  Question: "Every single row actually has the best result not in boldface, but.... the second best?"
>
> * Yes, they represent the second best because the MCTS(N=50) is the oracle results. Here we highlight the best results among the non-oracle versions.
>
> > Suggestion: "(Coulom, 2006) --> (Coulom, 2007)"
>
> * We find it should be (Coulom, 2006), and we check it out in Google Scholar.
>
> > Suggestion: "Descriptions of implementation details of MCTS / experiments in supplementary material are not sufficient."
>
> * We have added more details of our implementations in the revised version, including the mentioned MCTS-related hyperparameters, $C_{puct}$, $Q$-value assigned to unvisited children. Besides, we also add inputs/training/exploration/two-play MCTS four blocks in the "Training Details of Go" subsection in the Appendix. We think all these details are sufficient for the implementation of the board game Go.
>
> Finally, thanks for your suggestions. We have revised our paper and updated it on the website. And we highlight the changes and important details that reviewers have mentioned with blue color.

---

> > ### Comment · Reviewer_d2cw · 2021-11-18
> > **Satisfied with response by authors**
> >
> > Thank you for your responses. I have updated my score accordingly.
> >
> > Just to come back to this one (really minor, I realise that I'm nitpicking) issue:
> >
> > > We find it should be (Coulom, 2006), and we check it out in Google Scholar.
> >
> > Unfortunately Google Scholar is often wrong (and indeed many people incorrectly cite this particular paper). For example, Google Scholar also says that my Bachelor thesis should be cited as a Ph.D. thesis, but I'm fairly confident that that's incorrect too...
> >
> > You can find how the publisher itself (Springer) says that it should be cited here: https://link.springer.com/chapter/10.1007/978-3-540-75538-8_7#citeas
> >
> > The confusion in this case is probably because the conference was in 2006, but the proceedings of the conference only got published in 2007, and that's the date that matters in a citation.

---

> > > ### Author Response · Authors · 2021-11-19
> > > **Response to Reviewer d2cw**
> > >
> > > Thank you for your updates and explanations!
> > >
> > > We agree that it should be "(Coulom, 2007)" here and we will update this in the next revised version.
> > > Thank you again for your detailed explanations!

---

### Official Review · Reviewer_i2Lg · 2021-10-31

**Correctness:** 2
**Technical Novelty And Significance:** 2
**Empirical Novelty And Significance:** 2
**Recommendation:** 3
**Confidence:** 5

**Main Review:**

The paper incorrectly references the original MCTS/UCT version of Kocsis and Szepesvari for AlphaZero, however, AlphaZero used Rosin's P-UCT, not the original UCT.
The proposed method uses forward pruning/early termination in UCT to reduce the search effort. In effect, it introduces a variant of the UCT selection rule.
This idea is a fruitful idea, that has been tried by Lorentz in 2015 (Early Playout Termination in MCTS, Advances in Computer Games 2015) and by Hsueh in 2016 in Theoretical Computer Science for An analysis for strength improvement of an MCTS-based program playing Chinese dark chess.
These variations of UCT indeed also go back to the work of Auer and Cesa Bianchi (2002) on UCB1, on which UCT was based, which they reference as their first reference.
This idea has been shown to indeed improve performance in certain situations.
The proposed approach appears to be a variation on these older experiments.

The authors have  not presented comparisons to these earlier works in non-deep learning environments. I would be very interested to learn of a comparison of their new approach to these earlier selection rules. I would consider such comparisons essential for considering the new rule for publication in ICLR.

There is one unresolved reference to a section in the paper.


**Summary Of The Paper:**

This paper proposes the Virtual MCTS (V- MCTS), a variant of MCTS that mimics the human behavior, and is 50% more sample efficient, by performing a type of forward pruning.
MCTS is characterized as a model-based reinforcement learning algorithm, that imagines what the future would look like, using terminology borrowed from Sutton in his description of Dyna, a model-based algorithm.
I agree with the planning part, but am less certain if I agree with the learning characterization of MCTS.
Often, MCTS is used inside a model-based approach as the planning component. MCTS is usually not regarded as a full model-based approach.

**Summary Of The Review:**

The paper presents an interesting new termination criterion for MCTS.
The work should include comparisons to other selection rules, and misses references to some of these.
Without these comparisons, it is unclear how substantive the contributions are, and I do not recommend acceptance.
The language of the paper must also be improved.

---

> ### Author Response · Authors · 2021-11-16
> **Response to Reviewer i2Lg**
>
> Thank you for your comments and advice! We hope the following address your concerns:
>
> > Comment: "The paper incorrectly references the original MCTS/UCT version of Kocsis and Szepesvari for AlphaZero"
>
> * We agree with you and we have committed the changes in the revised version: use "Rosin's P-UCT" here and correct those references. Thank you for your correction.
>
> As for the two works you mentioned, there are some significant differences between V-MCTS and Lorentz's work as well as Hsueh's work. They are not in the same direction.
>
> We have mentioned in our paper that  there are four stages of per search in MCTS: selection, expansion, evaluation, and backpropagation. Lorentz's work MCTS-EPT is a termination rule in the evaluation stage while our method is a termination rule of when to stop the search in the outer loop. For a target move, MCTS will search for $N$ times iteratively and each iteration contains the four stages. In AlphaGo and some older methods, people evaluate the node state through random playouts to the end to get a reward. Lorentz proposed MCTS-EPT to stop the random playouts early and use an evaluation function to assess win or loss, which is an improvement in the evaluation part of normal MCTS. But V-MCTS is to terminate the whole search iteration, not the evaluation stage. Furthermore, MCTS-EPT similar ideas have already been applied in the evaluation stage of AlphaGoZero, AlphaZero, and later variants (including ours), which evaluate the node state through evaluation networks instead of running playout to the end.
>
> As for Hsueh's work, they investigate four techniques of MCTS, where only early playout terminations (EPT) is related to the early termination skill. But as mentioned above, EPT is aimed at the evaluation stage but V-MCTS is aimed at the outer iteration. V-MCTS will stop the iteration of search and use virtual expansion to generate a near-oracle policy distribution.
>
> To sum up, Lorentz's work and Hsueh's work are to terminate the playout in the evaluation stage of each search iteration in MCTS, but our method is to terminate the search iteration in MCTS. And AlphaZero-style works including ours have already used an evaluation network to evaluate the node state rather than run playout to the end, which is similar to MCTS-EPT. We have added more descriptions of the difference between ours and your mentioned works in the related works of the revised version.
>
> Finally, thanks for your suggestions. We have revised our paper and updated it on the website. And we highlight the changes and important details that reviewers have mentioned with blue color.

---

### Official Review · Reviewer_xM7t · 2021-11-02

**Correctness:** 3
**Technical Novelty And Significance:** 2
**Empirical Novelty And Significance:** 2
**Recommendation:** 5
**Confidence:** 4

**Main Review:**

The main contribution of the paper is an early-termination rule for MCTS. While the experiments show that the proposed V-MCTS is more efficient on certain tasks, my main concern is that the termination rule might be trivial in practice (e.g., simply perform fewer queries) since it does not really depend on the values. Suppose we have a budget of $N$ nodes to query. Intuitively, Lemma 2 suggests than when we have already visited close-to-$N$ nodes, then the final policy $\pi$ is unlikely to change too much. Although this is true, I think this does not necessarily tell us more than what we already know: since we only have a few queries left, the change in $N(s,a)$ will be minor, and so will the change in $\pi$. Therefore, I don’t see many conceptual differences between V-MCTS and selecting a better total computational budget (the total number of queries). Although I believe there are some important differences between V-MCTS and MCTS with optimized computational budget (see details below), but base on the experiments I can’t evaluate whether this is significant.

Specifically, I believe V-MCTS can more efficiently prune queries for “simple” states, since for these states $\pi$ could remain almost unchanged during the search. In contrast, for “harder” states, $\pi$ could change drastically after certain “rewarding” states have been discovered. I have a strong feeling that V-MCTS works because of this reason, and I think this point makes perfect sense. However, base on the current experiments I cannot decide whether this is true. I think the paper could be improved if the authors can analyze this case better.

From a theoretical perspective, Lemmas 1 and 2 do not provide new insights to the problem. The bound in Lemma 2 is based on the simple fact that $N(s,a)$ cannot change too much when the remaining number of queries is small. Therefore, in my humble opinion, the paper should focus more on empirically explaining the termination rule.

**Summary Of The Paper:**

The paper proposes Virtual MCTS, an early-termination rule for MCTS to improve its efficiency. Roughly speaking, the termination rule prunes the search process when the final policy at the root node is unlikely to change by too much from the current policy. This strategy improves the efficiency of AlphaGoZero-style algorithms. Specifically, the authors showed that Virtual MCTS improves the learning efficiency on 9*9 Go.

**Summary Of The Review:**

Overall, I tend to vote for rejection since (i) the paper’s theoretical findings are rather trivial and (ii) the paper does not provide good insight on why the termination rule works, or on what kind of problem can it performs better. But I think the proposed termination rule follows my intuition and the paper could be improved if the authors provide more detailed analyses on that.

---

> ### Author Response · Authors · 2021-11-16
> **Response to Reviewer xM7t**
>
> Thank you for your comments and advice! We hope the following address your concerns:
>
> > Concern: "the paper does not provide good insight on why the termination rule works, or on what kind of problem can it performs better."
>
> We think the visualization of Figure 3 gives an intuition of how V-MCTS works. It seems that we need more descriptions and analysis for this figure. The $k$ in Figure 3 is the number of search times given $N=150$ maximum search times. Here we can find that $k$ varies in different states. Specifically, $k$ will be small if the state is "simple". For example, on the board game Go, the start states are usually not complex because there are only a few stones on the board. Therefore, the $k$ is rather small in the start states as there is no necessity to do more searches. However, in shuban (the closing stage of the game), the situations are much more complicated. In this stage, the $k$ is larger.
>
> Besides, in Figure 1(a), we find that simply turning the budget will have a higher performance drop. In Figure 1(a), we compare the performance of V-MCTS and MCTS with a smaller number of queries during evaluations against GnuGo. The blue point named MCTS(N=90) has the same FPS (same queries) as V-MCTS(eps=0.1) while the performance is worse. This also indicates that simply changing the computational budget cannot obtain comparable results.
>
> In conclusion, V-MCTS can adjust the number of queries conditioned on the different situations of the states. And we think the two figures we mention above can provide insight on why the termination rule works.
>
> Finally, thanks for your suggestions. We have revised our paper and updated it on the website. And we highlight the changes and important details that reviewers have mentioned with blue color.

---

### Decision · Program_Chairs · 2022-01-20

**Decision:**

Reject

**Comment:**

The paper proposes Virtual MCTS, an early-termination rule for MCTS to improve its efficiency.
The basic idea is to introduce a termination rule that prunes the search process when the final policy at the root node is unlikely to change from the current one. The proposed approach is empirically evaluated on 9x9 Go and Atari games.

After reading the authors' feedback, all reviewers participated in the discussion without reaching a consensus.
Although all reviewers appreciated the authors' answers to their concerns, only one reviewer voted for acceptance. The other two reviewers,  while acknowledging some merits, still have concerns: the technical contribution is minor, the theoretical findings are quite trivial, it is unclear when the proposed termination strategy is could help.
In summary, this paper is borderline and I think it still needs some work to clearly break the bar of a top conference.